# Locked Intramedullary Nailing versus Compression Plating for Stable Ulna Fractures: A Comparative Study

**DOI:** 10.3390/jfmk6020046

**Published:** 2021-05-26

**Authors:** Vito Pavone, Marco Ganci, Giacomo Papotto, Giuseppe Mobilia, Umberto Sueri, Alpesh Kothari, Andrea Vescio, Gianluca Testa

**Affiliations:** 1Department of General Surgery and Medical Surgical Specialties, Section of Orthopaedics and Traumatology, University Hospital Policlinico “Rodolico-San Marco”, University of Catania, 95123 Catania, Italy; mrcganci@gmail.com (M.G.); giacomopapotto@gmail.com (G.P.); mobiliagiuseppe87@gmail.com (G.M.); umbertosueri@gmail.com (U.S.); andreavescio88@gmail.com (A.V.); gianpavel@hotmail.com (G.T.); 2Nuffield Department of Orthopaedics, Rheumatology and Musculoskeletal Sciences, University of Oxford, Oxford OX3 7LD, UK; alpesh.kothari@ndorms.ox.ac.uk

**Keywords:** ulna fracture, nightstick fracture, open reduction and internal fixation, ORIF, intramedullary nail

## Abstract

Background: Isolated ulna shaft fractures (USFs) are a relatively uncommon, but significant, injury. For unstable USF treatment, open reduction and internal fixation (ORIF) is the gold standard, while for stable USFs several procedures were described. The aim of this study is to compare the outcomes in patients with stable USFs treated by either ORIF or intramedullary nail (IMN). Methods: According to their surgical treatment, 23 eligible USF-affected patients were divided into ORIF (14 subjects) and IMN (nine subjects) groups. The subjects underwent postoperative clinical follow-up at 1, 3, 6, and 12 months, which included calculation of the Disabilities of the Arm, Shoulder and Hand (DASH) score and radiological follow-up. Time to union, time to return to sporting and occupational activities, duration of physical therapy, and surgical complications were recorded. Results: DASH scores improved in both groups at the 6-month follow-up (*p* < 0.001). The IMN cohort recorded better DASH scores at the 1- and 3-month follow-ups, while similar results were reported at the 6- and 12-month follow-ups. Earlier fracture union (*p* = 0.001) and return to sporting activities and work (*p* = 0.002) were seen in the IMN group, compared with the ORIF group. No complications were observed in the IMN group. Conclusions: The surgical treatment of isolated USF results in excellent functional and radiographic outcomes. IMN may be preferable, compared with ORIF, due to its faster recovery time, expedited union, and reduced likelihood of complications.

## 1. Introduction

Isolated ulna shaft fractures are a relatively uncommon, but significant, injury with an estimated incidence of 0.2 cases per 1000 population, representing 14% of adult forearm fractures [1]. These fractures most often result from a direct trauma to the ulna as the arm is raised overhead to defend the individual from an impact, and are thus commonly known as nightstick fractures [2]. According to Sauder and Athwal [3], isolated ulna fractures can be broadly described as stable or unstable, dependent on whether or not the interosseous membrane (IOM) is intact. For unstable ulna shaft fractures, there is consensus that open reduction and internal fixation (ORIF) with a compression plate is the gold standard treatment [3]. However, debate continues regarding the appropriate management of stable fractures. Some authors advocate nonoperative management, but this approach is associated with increased risk of nonunion, malunion, and joint stiffness [4]. Surgical intervention is favoured by some clinicians and can take the form of ORIF with a compression plate, intramedullary Rush pins, K-wires, or locked intramedullary nails [3]. In clinical practice there has been a shift away from less invasive techniques, such as intramedullary fixation, due to concerns about ongoing fracture instability and nonunion. Indeed, early devices, which lacked rotational stability, had nonunion rates of more than 10% [5], which is comparable to nonoperatively managed cases [6,7,8]. Contemporary intramedullary nailing (IMN) systems now provide much improved fracture stability, which in theory could mitigate the risk of nonunion. Intramedullary nailing also has the potential benefit of a reduction in operative morbidity due to its less invasive nature and improved fracture biology, compared with ORIF, as the fracture site is left relatively undisturbed with no extra periosteal stripping. It was hypothesised that patients treated with IMN would have a faster recovery, improved union rates, and reduced complications compared with those undergoing ORIF.

The purpose of this study was to compare radiological and clinical outcomes in patients with stable ulna shaft fractures treated by either ORIF or IMN.

## 2. Materials and Methods

### 2.1. Demographic Data

A retrospective medical record review was undertaken on all patients undergoing surgical treatment of an ulna shaft fracture at a single institution over a three-year period (2015–2017). Study inclusion criteria were: (1) Confirmed diagnosis of stable shaft ulna fracture (stability defined as per criteria of Sauder and Athwal [3], i.e., displacement <50% and angulation <10° on AP or lateral radiographs); (2) Patients aged between 18 and 75 years old; (3) Surgical treatment with ORIF or IMN as described below; (4) Complete clinical and radiographic dataset. Study exclusion criteria were: (1) Polytraumatised patients with other associated fractures; (2) Patients younger than 18 years of age; (3) Monteggia fractures; (4) Open or pathological fractures; (5) Follow-up less than 12 months; (6) Incomplete follow-up data.

A cohort of 23 subjects were eligible for this study and divided into two groups based on surgical intervention. The first group was composed of 14 patients treated between January 2015 and May 2016 with ORIF utilizing small fragment limited contact dynamic compression plates (LC-DCP). The second group comprised nine patients treated between June 2016 and December 2017 with 3.0/3.6 mm intramedullary ulna nails (Acumed, Hillsboro, OR, USA). The patient’s group selection was performed according to chronological criteria. Patient demographics are summarised in Table 1.

Table 1: Is there any way to ascertain which was the dominant hand of the participants? Both pre- and postoperatively the dominant arm would be used more and would likely have sustained greater hypertrophy (pre) and strain (post) surgery.

Groups did not differ significantly in their demographics or fracture pattern (*p* > 0.05). All subjects understood the purpose of the study and provided written, informed consent before participation. The study was carried out in accordance with relevant guidelines and regulations. Both of the procedures were performed by the same expert surgeon.

### 2.2. Surgical Techniques

#### 2.2.1. Open Reduction and Internal Fixation (ORIF)

Patients were positioned supine with the forearm in pronation on a radiolucent operating table with use of a pneumatic tourniquet. A skin incision was made overlying the subcutaneous border of the ulna, along a line drawn between the tip of the olecranon process and the ulnar styloid process. The interval between flexor carpi ulnaris and extensor carpi ulnaris was developed to access the ulna shaft. Open reduction of the fracture was undertaken whilst minimizing the degree of periosteal stripping. A small fragment 3.5-millimetre LC-DCP plate of appropriate length was used to stabilize the fracture in a method appropriate to the fracture type and degree of fragmentation. All patients underwent 2–3 weeks of postoperative splint immobilisation (Figure 1).

#### 2.2.2. Intramedullary Nail (IMN) 

For intramedullary nailing, an Acumed 3.0–3.6 mm × 210–270 mm Ulna Rod (Acumed, Hillsboro, OR, USA) was used according to the published surgical technique guide. A 1–2 cm longitudinal skin incision was made along the tip of the olecranon. Sharp dissection was then performed through the subcutaneous tissues and the triceps tendon, taking care to avoid the ulnar nerve which sits medial to the olecranon. A 6.1 mm cortical awl was used to establish the implant insertion point, and a guidewire was inserted and advanced across the fracture site. Sequential reaming was then undertaken to allow optimal isthmic fit and working length of the implant. The ulna nail was then inserted into the canal and across the fracture site. This was distally locked using a 3.5-millimetre interlocking screw [9]. There was no postoperative immobilisation.

All study patients had a period of rehabilitation with a dedicated musculoskeletal physiotherapist, the duration of which was determined by the physiotherapist based on functional progression (Figure 2).

### 2.3. Outcome Measures

All patients received standardised follow-up, with clinical, functional, and radiological follow-up at 1, 3, 6, and 12 months postoperatively. Patients underwent forearm radiographs, in anteroposterior and orthogonal lateral views, preoperatively and at each time point. Union was defined both clinically, as the lack of pain or tenderness at the fracture site, and radiographically, as bridging bone seen at, at least, three out of four cortices on orthogonal views. Delayed union was defined as union occurring between 3 and 6 months after index surgery. Nonunion was defined as a failure to proceed to radiographic union after 6 months from surgery.

The primary outcome measure was the Disabilities of the Arm, Shoulder and Hand (DASH) score. The DASH score is determined from a validated self-report questionnaire with 30 items, intended to measure physical function and symptoms in patients with any of several musculoskeletal disorders of the upper limb [10].

Secondary outcome measures included time to union, time to return to sporting and occupational activities, duration of physical therapy, and surgical complications.

### 2.4. Statistical Analysis

Continuous data are presented as means and standard deviations, as appropriate. The mixed-design analysis of variance (ANOVA) test and Tukey–Kramer method were used to compare the primary and secondary outcome measurements postoperatively at 1, 3, 6, and 12 months. For mean duration of physical therapy and return to work/sport period, the Mann–Whitney test was performed. The chi-squared test was used to compare fracture healing time. The selected threshold for statistical significance was *p* < 0.05. All statistical analyses were performed using IBM SPSS statistics software (version 24.0).

## 3. Results

DASH scores were improved at 1, 3, 6, and 12 months postop in both the ORIF group (*p* < 0.01) and the IMN group (*p* = 0.01). DASH scores were significantly higher in the IMN group compared with the ORIF group at 1- and 3-months postop (71.2 ± 13.53 vs. 36.5 ± 19.9, *p* < 0.01; 57.8 ± 12.33 vs. 26.6 ± 18.3, *p* < 0.01, respectively). There were no significant differences in DASH scores at 6 or 12 months between the IMN and ORIF groups (22.0 ± 6.2 vs. 17.7 ± 2.8, *p* = 0.06; 5.2 ± 2.2 vs. 4.7 ± 1.7, *p* = 0.54, respectively) (Table 2).

The IMN group required significantly less postoperative physical therapy time compared with the ORIF groups (23 ± 13 days vs. 86 ± 17 days; *p* < 0.01). Patients in the IMN group returned to sporting activities and work significantly earlier that those in the ORIF group (*p* < 0.01).

Clinical and radiographic union was attained in all study patients by the 12-month follow-up. Fracture union was, however, achieved earlier overall in the IMN group compared with the ORIF group (*p* < 0.01), with there being three cases of delayed union in the ORIF group compared with none in the IMN group.

No complications were observed in the IMN group, however five patients in the ORIF group complained of mild pain at the surgical site. There was also one case of a superficial surgical site infection reported in the ORIF group; this was resolved uneventfully with oral antibiotics.

## 4. Discussion

To the best knowledge of the authors, this is the first study directly comparing the clinical and radiographic outcomes for isolated stable ulna shaft fractures treated with either IMN or ORIF. We hypothesised that patients treated with IMN would have a faster recovery, improved union rates, and reduced complications compared with those who underwent ORIF. The primary outcome measure for this study was the DASH score, and indeed patients in the IMN group had improved scores at 1 and 3 months compared with those in the ORIF group. This finding, combined with evidence of earlier return to sports and work and fewer physical therapy sessions, confirms the belief that IMN results in faster recover than ORIF. The faster recovery is presumably, in part, a surrogate marker of the quicker time to union in the IMN group. Another important observation was that no complications were observed in the IMN group, however almost a half of those in the ORIF group had complications, albeit mild ones.

Isolated ulna shaft fractures are a relatively rare fracture and caused by high-energy direct trauma. Stable ulna shaft fractures can be treated nonoperatively with a mean time to union, when achieved, of approximately 18 weeks [5]. However, concerns are highlighted in the literature that nonoperatively managed fractures may result in unacceptably high rates of nonunion, ranging from 2–11% [11,12,13]. This, combined with worries about malunion, elbow and wrist stiffness, as well as heterotopic ossification of the IOM and compartment syndrome, have led to an increasing prevalence of surgical intervention in such cases. There are no absolute indications for operative treatment of isolated fractures of the ulnar shaft. Studies have recommended operative fixation for fractures in the proximal third of the ulna, fractures with displacement as low as 25% to >50%, or angulation of the fracture in any plane as low as 8° to >15° [5]. Whilst numerous methods exist for surgical intervention, the favoured surgical intervention for isolated ulna shaft fractures, according to the literature, is ORIF with LC-DCP or equivalent [3]. Historically, intramedullary methods have been applied. However, the main reason for the progressive disuse of this modality was a high rate of nonunion [14]. The first-generation nail, introduced in 1913, and the second-generation, introduced by Sage in 1959 [15], did not achieve sufficient rotational stability, due to the lack of locking or compression features [16]. Open reduction and internal fixation has the advantage of axial and rotational rigidity, stable fixation, and anatomical reduction; however, disruption of the fracture haematoma may have negative effects on the union, and excessive soft tissue and/or periosteal stripping could further compound this problem [14]. The latest generation of these nails, such as the one used in this study, were designed in the context of extensive anatomic and biomechanical analysis testing to minimize the risk of malrotation and instability [14]. Intramedullary nailing potentially allows less micromotion of the fractures and distributes the forces more uniformly across the fracture gap, compared with the ORIF technique [17]. Intramedullary nail fixation may be considered as an alternative for select diaphyseal fractures, such as those with extensive soft-tissue injury or pathologic fractures. However, the indications are limited at this time, and many advocate excluding Galeazzi, Monteggia, and severely comminuted fractures [18]. There are favourable data on the use of locked IMN in the literature, with reports of excellent clinical outcomes in 88.6–100% of patients. Our findings echo this, with patients treated with IMN recording DASH scores of 5.2 ± 2.2 at final follow-up, which is slightly better than the DASH scores observed in the general population [19,20,21].

The critical findings of this study relate to the speed of recovery. The differences in DASH scores between IMN and ORIF only ceased to be significantly different at 6 months. The expedited recovery was further evidenced by the fact that the ORIF group had 3.7 times more physical therapy and took 2.5 times as long to return to work and sports. This has a profound impact on not only patient morbidity, but healthcare resource utilisation and societal burden. There are a number of potential reasons why IMN could result in faster overall recovery than ORIF for isolated ulna shaft fractures. As previously mentioned, the technique is less invasive than ORIF, and has been shown to have minimal bleeding, low infection rates, no heterotopic ossification of the IOM, and a quicker restoration of forearm and wrist range of motions [16,17]. The absence of initial cast immobilisation may have also been a factor, and is something that we should consider for all operatively managed fractures moving forward. A prerequisite for recovery is fracture union, and in concordance with the clinical and functional findings, fractures treated with IMN healed faster that those treated with ORIF. It is our belief that minimizing fracture haematoma and soft tissue disruption, as well as a favourable biomechanical environment, all have a role to play in our observations, along with the surgical factors. The importance of each specific factor is, however, beyond the scope of this study. Ultimately, it is worth considering that both surgical options evaluated in this study represent a good solution in the management of the stable ulna fracture. All patients eventually healed, attained a good functional outcome, and had an acceptable complication profile. Randomised controlled trials are strongly encouraged in order to obtain high quality findings.

There are several limitations inherent to the study design. Firstly, the study was retrospective in nature, however as per departmental protocol the data were collected prospectively and in a robust manner. The dominant limb was not recorded in the study, which could influence the preoperative hypertrophy and postoperative strain. It would have also been ideal to have a matched, nonoperatively managed group for further comparison to assess whether surgical intervention is preferable to nonoperative management. Finally, the sample size was small, and as such the study was underpowered to observe small to moderate effect sizes, increasing the risk of a type II error. That withstanding, this study serves to provide good pilot data from which future, appropriately powered, well-designed, prospective studies can stem.

## 5. Conclusions

In conclusion, surgical treatment of isolated stable ulna shaft fractures results in excellent functional and radiographic outcomes. Intramedullary nailing may be preferable to ORIF due to its faster recovery time, expedited union, and reduced likelihood of complications.

## Figures and Tables

**Figure 1 jfmk-06-00046-f001:**
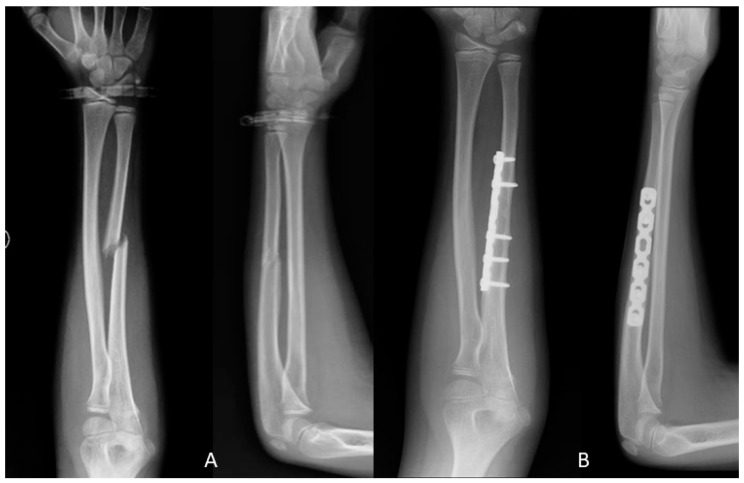
(**A**) Ulna fracture preoperative X-ray; (**B**) One year after ulna fracture open reduction and plating X-ray.

**Figure 2 jfmk-06-00046-f002:**
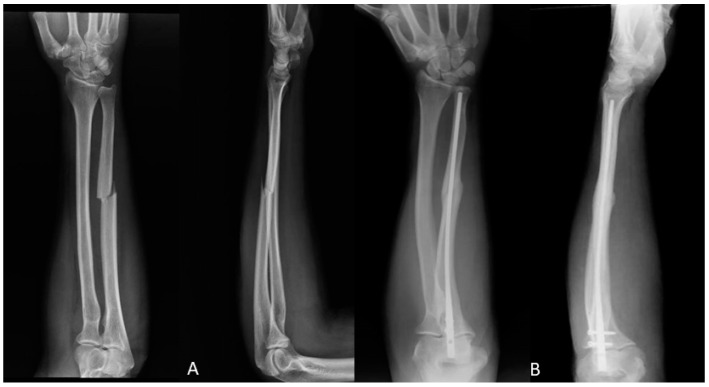
(**A**) Ulna fracture preoperative X-ray; (**B**) One year after ulna fracture intramedullary nailing X-ray.

**Table 1 jfmk-06-00046-t001:** Group demographics.

	ORIF GROUP	IMN GROUP	Total
Patients	14	9	23
Male (%)	10	6	16 (69.6%)
Female (%)	4	3	7 (30.4%)
Mean age (Range)	44.8 years (18–67 y)	47.2 years (22–83)	46.4 years (18–83)
Fracture side (%)	8 left	5 left	13 left (56.5%)
	6 right	4 right	10 right (43.5%)

**Table 2 jfmk-06-00046-t002:** Clinical and radiological assessment, including Disability of the Arm, Shoulder and Hand score, radiological healing, physical therapy, and return to work and sports activities.

		ORIF GROUP (14 Patients)	IMN GROUP (9 Patients)	*p*-Value (95% C.I.)
Mean DASH score	1 month	71.19 ± 13.53	36.54 ± 19.91	<0.01 (20.2–49.1)
	3 months	57.82 ± 12.33	26.57 ± 18.35	<0.01 (18.0–44.5)
	6 months	22.03 ± 6.23	17.67 ± 2.82	0.06 (−0.26–8.98)
	12 months	5.21 ± 2.21	4.68 ± 1.66	0.54 (−1.26–2.32)
Radiological healing	1 month	2	8	<0.01
	3 months	4	0	
	6 months	8	1	
	12 months	0	0	
Mean physical therapy (days)		85.6 ± 16.74	23.33 ± 13.22	<0.01 (48.5–76.0)
Return to work or sport (months)		5.8 ± 2.11	2.3 ± 2.59	<0.01 (1.45–5.54)

## Data Availability

The data presented in this study are available on request from the corresponding author. The data are not publicly available due to hospital confidentiality.

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
