# Peer review of "Locked Intramedullary Nailing versus Compression Plating for Stable Ulna Fractures: A Comparative Study"

_jfmk, 2021, doi:10.3390/jfmk6020046_

Round 1
Reviewer 1 Report
The authors compared the effects of open reduction and internal fixation (ORIF) and intramedullary nail (IMN) in fixing ulna shaft fractures. They showed that IMN increased the DASH score at 1 and 3 months, early healing and return to work and sport. Considering the retrospective nature of this study and its small sample size, I would like to raise several issues:
- Was sample size calculation performed before data collection?
- Table 1: The unit of the data should be presented. Are they mean and range, as well as number (percentage)?
- Given repeated data across three follow up period between two groups are measured, would the authors consider mixed-design ANOVA in analysing DASH to derive within and between-group difference?
- For the difference in DASH at 1 month and 3 months, are the p-values <0.0001 or =0.0001?
- I would definitely like to see some example of radiographic pictures.
- 2nd paragraph of the discussion is too long. I suggest breaking it up into several smaller paragraphs.
Author Response
The authors compared the effects of open reduction and internal fixation (ORIF) and intramedullary nail (IMN) in fixing ulna shaft fractures. They showed that IMN increased the DASH score at 1 and 3 months, early healing and return to work and sport. Considering the retrospective nature of this study and its small sample size, I would like to raise several issues:
Q1) Was sample size calculation performed before data collection?
A1) No sample size calculation was performed. The missing statistical analysis was discussed in limits of the study
Q2) Table 1: The unit of the data should be presented. Are they mean and range, as well as number (percentage)?
A2) Requested modifies were performed.
Q3) Given repeated data across three follow up period between two groups are measured, would the authors consider mixed-design ANOVA in analysing DASH to derive within and between-group difference?
A3) the ANOVA use was specified in the method section
Q4) For the difference in DASH at 1 month and 3 months, are the p-values <0.0001 or =0.0001?
A4) the p values are =0.0001, the typos were corrected.
Q5) I would definitely like to see some example of radiographic pictures.
A5) Pre- and 1 year post-operative Xrays were added
Reviewer 2 Report
Details:
Line 18: Is it UF or USF here?
Line 21: Is the word „score“ missing here?
Line 27: What is SUSF? USF has been defined, but SUSF not.
Line 43 ff: „As such….“ – is this sentence grammatically complete?
Line 71 ff: I suspect a selectional bias here: how were the patients allocated to the treatment method? Was it depending on the fracture details? Or was it depending on the surgeon in charge? Clarifying the allocation details would improve the quality of the study.
Table 1: “Mean age: 44.8 (18-67)” – The numbers in the brackets, are they minimum/maximum?
“Total male” – 16/23 is 69.6%, not 69.5; Total female 7/23 is 30.4%, not 30.5.
Line 229: Maybe the selectional bias in allocation the patients to one of the two treatment methods has to be mentioned in the section about the limitations of the study.
Thank you for the opportunity to review this paper.
Author Response
Q1) Line 18: Is it UF or USF here?
A1) The typo was corrected.
Q2) Is the word „score“ missing here?
A2) The typo was corrected.
Q3) What is SUSF? USF has been defined, but SUSF not.
A3) The typo was corrected.
Q4) As such….“ – is this sentence grammatically complete?
A4) the grammar errors were corrected
Q5) I suspect a selectional bias here: how were the patients allocated to the treatment method? Was it depending on the fracture details? Or was it depending on the surgeon in charge? Clarifying the allocation details would improve the quality of the study. Line 229: Maybe the selectional bias in allocation the patients to one of the two treatment methods has to be mentioned in the section about the limitations of the study.
A5) The treatment choice was performed according to chronological criteria. Additional information were added in the text
Q6) Table 1: “Mean age: 44.8 (18-67)” – The numbers in the brackets, are they minimum/maximum? “Total male” – 16/23 is 69.6%, not 69.5; Total female 7/23 is 30.4%, not 30.5.
A6) Requested modifies were performed.
Thank you for the opportunity to review this paper
Round 2
Reviewer 1 Report
I request the authors to revisit their statistical analysis with the help of a statistician.
For this previous comment: Given repeated data across three follow up period between two groups are measured, would the authors consider mixed-design ANOVA in analysing DASH to derive within and between-group difference?
The authors responded: the ANOVA use as specified in the method section.
Further comment: Which TYPE of ANOVA were the authors using to analyse the data? Due to the time X group design (for DASH score), I am suggesting the use of MIXED-DESIGN ANOVA or UNIVARIATE analysis with time and group as independent variables. For other parameters (mean physical therapy period and return to work/sports), there are only two groups, the authors should use student's T-test or Mann-Whitney test.
I also question the need to present p-values up to 4 decimal places.
Author Response
Q1) I request the authors to revisit their statistical analysis with the help of a statistician.
For this previous comment: Given repeated data across three follow up period between two groups are measured, would the authors consider mixed-design ANOVA in analysing DASH to derive within and between-group difference?
The authors responded: the ANOVA use as specified in the method section.
Further comment: Which TYPE of ANOVA were the authors using to analyse the data? Due to the time X group design (for DASH score), I am suggesting the use of MIXED-DESIGN ANOVA or UNIVARIATE analysis with time and group as independent variables. For other parameters (mean physical therapy period and return to work/sports), there are only two groups, the authors should use student's T-test or Mann-Whitney test.
A1) Thank for your suggestion. The MIXED-DESIGN ANOVA was performed for Dash score analysis, while for mean physical therapy and return to work/sports periods Mann-Whitney test was used. The p-values did not change after the further analysis.
Q2) I also question the need to present p-values up to 4 decimal places.
A2) p-values were reported according to your suggestion.
Round 3
Reviewer 1 Report
The authors have addressed my previous comments.
Author Response
Thank you for your reply